# Investigation of Machine Learning Techniques for Disruption Prediction Using JET Data

**Joost Croonen, Jorge Amaya** and **Giovanni Lapenta***

Centre for Mathematical Plasma Astrophysics, KU Leuven, 3000 Leuven, Belgium
* Correspondence: giovanni.lapenta@kuleuven.be

**Abstract:** Disruption prediction and mitigation is of key importance in the development of sustainable tokamak reactors. Machine learning has become a key tool in this endeavour. In this paper, multiple machine learning models are tested and compared. A focus has been placed on the analysis of a transition to dimensionless input quantities. The methods used in this paper are the support vector machine, two-tiered support vector machine, random forest, gradient-boosted trees and long-short term memory. The performance between different models is remarkably similar, with the support vector machine attaining a slightly better accuracy score. The similarity could indicate issues with the dataset, but further study is required to confirm this. Both the two-tiered model and long-short term memory performed below expectations. The former could be attributed to an implementation which did not allow error propagation between tiers. The latter could be attributed to high noise and low frequency of the input signals. Dimensionless models experienced an expected decrease in performance, caused by a loss of information in the conversion. However, random forest and gradient boosted trees experienced a significantly lower decrease, making them more suitable for dimensionless predictors. From the disruption detection times, it was concluded that several disruptions could be predicted at more than 600 ms in advance. A feature importance study using the random forest indicated the negative impact of high noise and missing data in the database, suggesting improvements in data preparation for future work and the potential reevaluation of some of the selected portable features due to poor performance.

**Keywords:** machine learning; plasma physics; tokamak; disruptions; portable prediction models; SVM; gradient-boosted trees; random forest; LSTM; JET

## 1. Introduction

Disruptions are energetic events in tokamak fusion devices resulting from the loss of plasma confinement, which leads to severe thermal and mechanical stress on device components [1,2]. In large scale devices, this can cause lasting damage to the reactor, in particular but not limited to plasma facing components [3]. Disruptions, therefore, pose a serious problem in the development of tokamak reactors, and avoiding them is a high priority goal in the further development of the technology [4].

Boozer [1] describes two main factors as the cause for plasma disruptions, namely, the robustness of the plasma centering in the confinement chamber, and the robustness of the magnetic surfaces containing the plasma. This conclusion is based on a study of JET disruptions [5]. The breaking of magnetic surface robustness is further attributed to several effects: (1) A sensitivity to the profile of the net parallel current when the Greenwald density limit is exceeded or when rapid plasma cooling occurs and the profile of the pressure gradient is influenced by a strong internal transport barrier. (2) A sensitivity to the bootstrap current due to its ability to widen magnetic islands causing a breakup of the magnetic surface. (3) Changes in the plasma rotation and the accompanying decreases in their magnetic surface, preserving the effects that decrease island formation. (4) The negative influence of external currents due to the effects of error fields and resistive

wall modes. The mitigation of disruptions is by a controlled rapid decrease in the plasma energy prior to a disruption. This can be achieved in several ways; for example, by introducing cold massive particles into the plasma via a cold gas injection or pellet injection, which will rapidly absorb and radiate the energy away. Both methods were studied for use in the ITER (International Thermonuclear Experimental Reactor) [6–8]. For these mitigation methods to work, it is, however, imperative that these disruptive events can be predicted in advance of a non-trivial task, so that they can be activated in time.

In the last two decades, machine learning has become a promising tool in overcoming this problem. These methods do not rely on a first principle understanding of the plasma and disruption physics. Instead, they use the vast amount of experimental data available to train models to detect patterns associated with disruptions. Such models have been designed for several devices around the world such as JET (Joint European Torus) [9,10], ASDEX-Upgrade (Axially Symmetric Divertor Experiment) [11], J-TEXT (Joint Texas Experimental Tokamak) [12], DIII-D [13] and EAST (Experimental Advanced Superconducting Tokamak) [14], amongst others. In recent work, other advanced learning algorithms are being explored in work such as Kates-Harbeck et al. [15], which discusses the use of deep recurrent neural networks as a predictor for JET, and Pau et al. [16], in which a generative topographic mapping method is discussed to identify disruptive boundaries in the JET. Vega et al. [3] gives an overview of several ML techniques used for disruption prediction and Zhu et al. [17] discusses the use of a hybrid deep learning model for general disruption prediction. In this paper, based on Croonen [18], a comparative study is performed between different machine learning methods applied to JET data: support vector machines [19], two-tiered support vector machines [9], random forest [20], gradient boosted trees [21] and long-short term memory [22]. The models were designed to facilitate this comparison, and the goal was not to build the best performing predictor. This analysis gives an insight into the benefits and drawbacks of each individual method. Portability is an important property for disruption prediction models, as it describes how easy it is to apply a model to data from a different device. This is particularly useful for the next generation of tokamaks, such as ITER, since there are no experimental data available to train predictor models for such devices. An important but not sufficient condition for portability is a transition to dimensionless quantities. They allow for the quantities to be compared between devices. However, these quantities do not comprehensively encode all the disruption information. Therefore, the disruptive boundaries can differ between devices. The effects of this on portable disruption predictors, though very important, are outside the scope of this paper. This transition to dimensionless quantities has been analysed for the different models in this paper, though a fully exhaustive analysis on the portability of the methods is not present.

The following topics will be discussed in this paper: Section 2 discusses the data and normalisation techniques that were used for the training of the models. Section 3 describes the machine learning methods used in this work. Section 4 discusses the performance metrics and results of the ML models. Section 5 summarises the most important results of this paper. The code used to train the models and analyse the performance is available at https://github.com/JoostCroonen/ML_Tokamak_Disruption_Prediction, (accessed on 4 December 2022).

## 2. Data

The different Machine Learning (ML) algorithms were trained using data from JET, governed by the Culham Centre for Fusion Energy (CCFE). Disruption events in this dataset are recorded in the ITPA (international tokamak physics activity) disruption database (IDDB) [23]. IDDB contains shot numbers and times of disruption for 1170 disruptive events in JET, none of which were artificially induced. The disruption range between shots 32,157 and 79,831, and ranges over several carbon wall campaigns. Throughout this period, JET underwent several major changes. The impact of these changes on the data has not been taken into account, since all data were treated identically. This approach was chosen for its simplicity but has a negative impact on the model performances. This was deemed

acceptable, since the focus in this paper lies on a comparative study between models and no applicable predictive capabilities are claimed.

The input signals for the ML models, also called features, are listed in Table 1. We use the features from the work by Rattá et al. [24] and Rea et al. [13]. In the former, a genetic learning algorithm is used to select the best features, while in the latter, a univariate statistical analysis of the potential features was performed. Since we use the same features, these analyses were not repeated in this study. Any features outside of the selection have not been considered, as their importance is deemed low given their exclusion by both of these papers. Several of the features identified by these papers have been discussed in papers such as Pustovitov and Ryabushev [25] and Schlisio [26] to analyse their impact on fusion plasmas. All features were normalised around 0 with a standard deviation of 1. The sampling frequency of all the signals was set to 30 ms, which is the frequency of the majority of the signals. Some signals are unavailable for specific shots, which will be referred to as gaps in the database. The data availability is listed in Table 1. The incomplete dataset makes it more difficult for ML models to learn from the available data. In particular, the neutral beam injector (NBI) power and ion-cyclotron radio heating (ICRH) power are frequently missing. Due to the relatively small dataset, it was decided that it was best to keep all shots, even if data were missing from them. The missing data were set to zero. This is a reasonable assumption for the input powers, as the most frequently missing signals, since the absence of the data suggests that the corresponding heating devices were offline, though this was not confirmed. The choice of retaining all the features is questionable, in particular $P_{ICRH}$, since it is frequently missing. The rationale for it is two fold. Firstly, it is undesirable to reject shots with missing features, since the amount of available shots is low and this would reduce it even further. Secondly, we can a priori expect a feature that is often missing (set to zero for missing entries) and will have a small weight in the training process. This is in fact confirmed by the results (see Figure 1 below). This allows us to maintain all the features as described in Rattá et al. [24] and Rea et al. [13] and will enable us to use them to create a set of dimensionless features, as described in Section 3.

**Table 1.** List of the input features used to train the different predictor models. Selection based on Rattá et al. [24] and Rea et al. [13]. $\sigma$ is the standard deviation and $\mu$ is the mean. The last column represents the relative standard deviation.

| Name | Description | Data Availability | $\frac{\sigma}{\mu}$ |
|------|-------------|-------------------|----------------------|
| $I_{pla}$ | Plasma current $[A]$ | 100% | 0.83 |
| $MLA$ | Mode lock amplitude $[T]$ | 100% | 2.19 |
| $l_i$ | Plasma internal inductance | 81.3% | 0.90 |
| $W_{dia}$ | Diamagnetic energy $[J]$ | 99.0% | 1.35 |
| $\dot{W}_{dia}$ | Time derivative of the diamagnetic energy $[W]$ | 99.0% | 64.37 |
| $n_e$ | Electron density $[m^{-3}]$ | 99.7% | 1.17 |
| $P_{out}$ | Radiated output power $[W]$ | 99.7% | 1.80 |
| $P_{NBI}$ | Neutral beam injection input power $[W]$ | 70.0% | 1.75 |
| $P_{ICRH}$ | Ion cyclotron radio heating input power $[W]$ | 42.5% | 3.05 |
| $q_{95}$ | Edge safety factor | 100% | 1.08 |
| $B_{\phi}$ | Toroidal magnetic field strength $[T]$ | 100% | 0.81 |

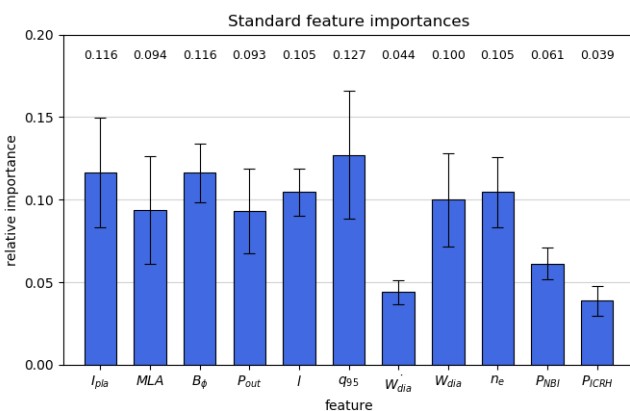

**Figure 1.** Relative importance of the different standard input features, as obtained from the random forest model.

From each shot, three 600 ms sequences (equivalent to 20 datapoints) were extracted as samples. Disruptive (positive) samples are taken from the range of 30–630 ms before disruption. The final 30 ms is left as the reaction time for disruption mitigation systems. Non-disruptive samples (negative) are taken from 1000 ms–1600 ms and 4000 ms–4600 ms before disruption. No non-disruptive shots were used during training due to the lack of a dataset of clean pre-labelled shots of this type. This approach relies on the assumption that at sufficiently long timescales before a disruption, the disruptive nature of a shot should not yet be discernible. This assumption has also been reported by [13]. Since all the shots used are disruptive, they inherently operate close to disruptive limits. This limits the models to only distinguishing disruptive and non-disruptive shots near these limits, as it was not trained on any data far from them. However, this is also the area where the distinction between disruptive and non-disruptive shots would be most difficult and therefore provides a good worst case test to differentiate the ML methods. Nonetheless, the performance values mentioned throughout this paper will therefore only apply to data which are close to these disruptive boundaries.

The timescales on which a shot becomes disruptive can vary significantly, but this variance was not taken into account here. Instead, it was fixed to 600 ms. This particular choice is arbitrary. The value was chosen to make sure the non disrupting data sets are truly and safely non disrupting. From more recent studies [16], 600 ms is deemed a very safe choice. Perhaps too safe: several studies [9,27] have used significantly shorter intervals. The consequence of this conservative choice makes the disruptive shots longer and, therefore, the actual disrupting samples are diluted by mislabelled stable periods in the shot. Though this has obvious negative effects on the performance of the models, it was deemed acceptable in this study, which does not aim at creating an operational tool but rather aims at comparing the different methods. In fact, by making the detection of disruptions more difficult, we are making a more stringent stress test of the different methods compared. In future work, a detailed analysis of the effect of this disruptive window on the different models should be considered. It can be expected that the disruptive window would be different for each disruption [27]. Therefore, a static disruptive window would always cause some mislabelling of the samples. Taking the varying disruptive windows into account could significantly improve the predictive capabilities of all ML methods. However, the identification and labelling of these disruptive windows is complex and time consuming.

The dataset is split into a training set (60%), which is used to train the models; a test set (20%) to determine the performance; and a validation set (20%), which is used to determine the hyperparameters (see Section 3). The splitting is conducted randomly per sample; thus, different samples from the same shot are not guaranteed to be in the same subset. Extending the dataset with new data could allow the analysis of the impact of this choice, and the refinement of the models.

## 3. Methods

ML is a useful tool for the prediction of disruption. It does not require a thorough understanding of the mechanisms behind these events. ML makes use of the large amount of experimental data that are available from experimental reactors, to detect patterns associated with disruptions. This information is used to classify new samples as either disruptive or non-disruptive. Though no physical understanding is necessary to use ML, it can still be very valuable to consider the physics when setting certain parameters and choosing input features.

ML algorithms non-linearly process the given inputs using a large number of parameters which determine the output. To train the models, the error between the calculated output and the desired output is calculated using a cost function, which is subsequently minimised.

Each of the algorithms used in this work has a set of hyperparameters determining their learning behaviour. These have been optimised for the validation set through a grid search method. The performance values are measured using the test set. The validation set is distinct from the test set to avoid tailor-fitting the hyperparameters to the test set, which would unrealistically inflate the performance results.

All the models were trained twice: once using the standard inputs previously described in Table 1 and once using dimensionless inputs, which are listed in Table 2. The latter were chosen based on Tang et al. [28] and Rea et al. [13]. Dimensionless inputs are of particular interest since they are device independent. This makes them ideal to use in portable predictors; models which can be used across devices. This means they can be trained on and applied to different devices. Note that a transition to dimensionless quantities is not sufficient to ensure good portability. An important problem in the issue of portability is the change in the relative disruptive domains in the feature space of different devices. This has not been studied in this paper, but it is vitally important in designing portable models [29].

The code for training and analysing the different models is available at https://github.com/JoostCroonen/ML_Tokamak_Disruption_Prediction (accessed date 4 December 2022).

**Table 2.** List of machine independent features that are used to train the dimensionless predictors and how they are derived from the original set of features in Table 1.

| Name | Description | Formula |
|------|-------------|---------|
| $I_N$ | Normalized plasma current | $I_{pla}/aB_\phi$ |
| $f_{MLA}$ | Mode lock amplitude fraction | $MLA/(B_\phi)$ |
| $l_i$ | Plasma internal inductance | $l_i$ |
| $f_{gw}$ | Greenwald density fraction | $n_e/(I_{pla}/\pi a^2)$ |
| $f_P$ | Radiated power fraction | $P_{out}/(P_{NBI} + P_{ICRH} - \dot{W}_{dia})$ |
| $q_{95}$ | Edge safety factor | $q_{95}$ |

### 3.1. Support Vector Machines

The first algorithm is the support vector machine (SVM). It is designed to find the hyperplane that best separates two distinct classes in the feature space. It decides on the best hyperplane by maximising the margin around it. When a dataset cannot be linearly separated, a transformation function can be used to map the data into a higher dimensional space where the datapoints are linearly separable, which is known as the kernel trick. A detailed description and mathematical derivation of this model can be found in Hastie et al. [30] (Chapter 12).

The simplicity of the model and its low computational cost make it an excellent tool for quick iteration and testing. It has also been used in previous work, such as Moreno et al. [9]. It was implemented using the python package Scikit-learn [31]. It uses the non-linear radial basis function (RBF) as a kernel with regularisation parameter $C = 0.33$ and kernel size $\gamma = 0.1$. Based on Moreno et al. [9], a 2-tiered SVM classifier was also designed. It works based on a sliding window scheme where the first tier SVMs will observe 3 subsequent timesteps. The results of the 3 first tiers are then used by a second tier SVM, which outputs

the final classification. A schematic of this is shown in Figure 2. The first tier is identical to the standard SVM, except that it outputs probabilities rather than discrete classification, while the second tier uses a linear kernel and $C = 0.01$. Due to the implementation in Scikit-learn, the two tiers were trained independently. This means that errors could not propagate from the second into the first tier SVMs. This shortcoming is expected to have a negative impact in the performance of this model.

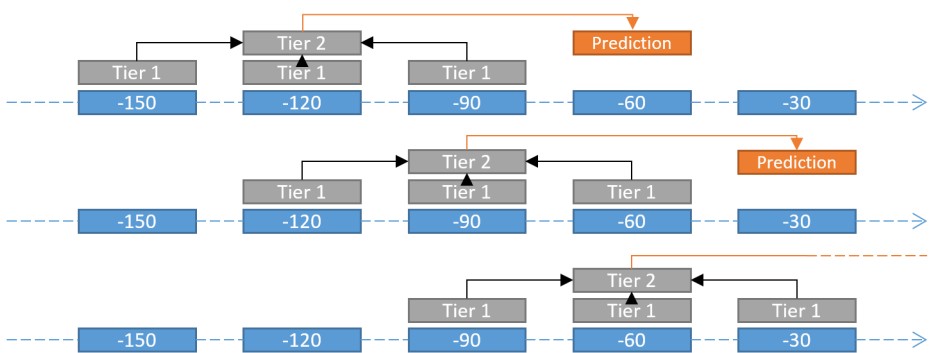

**Figure 2.** Schematic of the sliding window scheme used by the 2-tiered SVM. (Source: Croonen [18]).

### *3.2. Random Forest*

Random forest (RF) solves classification problems using many different decision trees (DT). DTs are tree-like data structures where each branch point represents a decision point. There, one or more variables can be evaluated and based on the outcome; the process will continue along the corresponding branch to the next branch point. This continues until it reaches one of the end points of the tree, known as the leaves. However, singular DTs are not well suited to represent the behaviour of the disruptions. DTs are characterised by a high variance error and are thus prone to overfitting. This can be overcome by combining the outputs of many DTs through a majority voting system, which makes for a much more robust classification model. The influence of each DT towards the final decision is adjusted during the training process based on their individual performance. A detailed discussion about RF and their training can be found in Hastie et al. [30] (Chapter 15). RF was also implemented using Scikit-learn. It used 1000 initial DTs without a maximum tree depth and bootstrap set to true. As maximum features per split, it used 3 and 2, respectively, for the standard and dimensionless model. The minimum samples per leaf and minimum samples per split were 1 and 2, respectively. The RF method allows for the analysis of the relative importance of the different inputs. A discussion on how this works can be found in Hastie et al. [30] (pp. 367–369, 593). Such an analysis is discussed in Section 4.3.

### *3.3. Gradient-Boosted Trees*

Boosting algorithms are a group of iterative approximation techniques. On each iteration, a new function is introduced, whose parameters are adjusted to approximate the residual error from the previous step. In the case of gradient-boosted trees (GBT), the estimator functions in each step are DTs. On the first step, the residual error is equal to the value of the expected outputs. As the algorithm progresses, it will approximate the desired output because each DT corrects for the error from the previous step. More details on the working and training of this method can be found in Hastie et al. [30] (pp. 342–343). This method was also implemented with Scikit-learn and used 1000 iterations with trees of maximum depth 3, a learning rate of 0.1 and minimum samples per split and per node of 2 and 1, respectively.

### *3.4. Long Short-Term Memory*

Long short-term memory (LSTM) is a variation of a recurrent neural network (RNN). A RNN is a neural network (NN), taking as an additional input the output signals of the

NN at the previous timestep. This ensures that there is a link to past events uncovering time dependent behaviour. LSTM improves on this algorithm by introducing a 'memory' called the cell-state. Only the read and write operations are applied to this cell-state during training, allowing the NN to maintain information over a long time.

A more detailed discussion on the working and training of LSTM can be found in Goodfellow et al. [32] (pp. 410–411). The LSTM implementation used in this work is a many-to-one network. It works like a sliding window scheme. On each timestep, the algorithm looks at the last 20 timesteps, on which it performs the LSTM algorithm to make a prediction. This is schematically shown in Figure 3. This method ensures good generalisability because it is agnostic of the total sequence length, which can vary significantly between shots. However, it is limited to the information of the last 20 timesteps, corresponding to a 600 ms time period. The implementation of this method was conducted using pytorch [33]. Based on a hyperparameter search, the standard model used a two layer LSTM with 22 nodes, while the dimensionless model used a one-layer LSTM with 18 nodes.

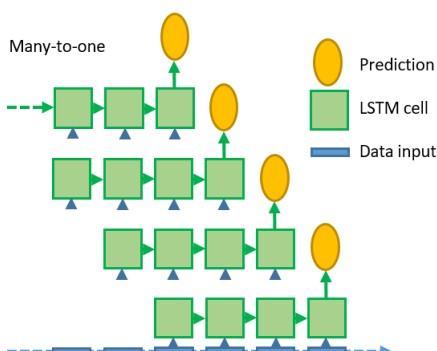

**Figure 3.** Schematic of the sliding many-to-one implementation of LSTM. (Source: Croonen [18]).

A single fully connected layer was used to reduce the final LSTM output to a single prediction value. The model was trained using the ADAM optimiser [34] and using a binary cross-entropy error function. It was trained for 1000 cycles on a single GPU using stochastic mini-batching to reduce the memory footprint.

## 4. Results

Commonly used metrics in the field of ML are used to discuss the performance of the models in this work. In a predictive model with binary selection, the recall ($R$) is the ratio of the true positives (tp, i.e., the samples that were correctly identified as positive) over the actual positives (ap, i.e., the samples that are actual disruptions). The precision ($P$) is the ratio of the true positives over the predicted positives (pp, i.e., the samples which are predicted by the model, correctly or incorrectly, to be positive).

$$R = \frac{tp}{ap} \tag{1}$$

$$P = \frac{tp}{pp} \tag{2}$$

The recall is therefore an estimate of the fraction of disruptions that have been correctly predicted by the model. The precision estimates the chance that a prediction is correct when it predicts a disruption. Ideally, both should be close to one. A combined metric, called the F1-score, mixes these two metrics:

$$F1 = \frac{2RP}{R + P} \tag{3}$$

Because of the severe consequences of disruptive events, it could be advantageous to prioritise the optimisation of the recall. However, a low precision would cause disruption mitigation techniques to be activated unnecessarily, decreasing the performance of the fusion device. To avoid a bias in this work, the two metrics were chosen to be equally important, though it is recognised that changing this balance could help fine-tune the performance of the models to better suit a particular requirement. This choice ensures that any bias observed in the results (see Figure 4) is solely caused by the models themselves. A sequence of 600 ms is labelled positive if at least one timestep is classified as disruptive and negative otherwise. A predicted positive is classified as a true positive if a disruption is imminent within 600 ms, i.e., if it is in the sequence before a disruption.

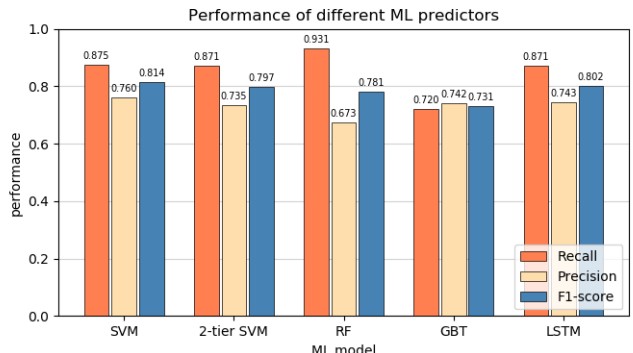

**Figure 4.** Recall, precision and F1-score of the different standard prediction models.

### 4.1. Predictor Performance

The performance metrics of the different techniques are shown in Figure 4. It is apparent that most algorithms prefer optimising the recall. This suggests that for this dataset it might, in general, be easier to avoid false negatives compared to false positives. RF has the strongest imbalance towards the recall while GBTs are the exception, with a very balanced performance instead. The F1 scores of the different models are very similar. SVM has the best overall performance, while GBT has the lowest. The low variance between the models could indicate that one or more of the assumptions in the design of the dataset, as discussed in Section 2, severely complicates the classification of a subset of the data for all models. For example, this could be due to some data dating from before a major update in JET, or some disruptions developing on timescales much shorter than 600 ms, which causes many of their datapoints to be effectively mislabelled. The frequent missing data could also cause certain shots to become hard to identify for all models. Moreno et al. [35] suggests that the plasma of internal inductance plays an important role in disruption prediction, while this information is missing in 18.7% of shots. Rattá et al. [36] shows that using learning-based regression models to fill in the gaps in data can significantly improve performance. They considered symbolic regression, based on genetic programming, and support vector regression models. Alternatively, a k-nearest-neighbour method or similar data imputation methods can be used to replace the missing data [37]. Careful analysis of the incorrectly predicted shots could provide insight into their causes in future work. The use of generated data could also be used to extend the dataset to further analyse the performance results. Interestingly, the 2-tiered SVM does not outperform the standard SVM. Considering that the first tier in this model is nearly identical to the standard SVM, it is remarkable that the model shows the worst performance. As suggested in Section 3.1 in the discussion on SVMs, using a two-tiered SVM which properly propagates errors between tiers during the training could provide noticeable performance improvements, as the two tiers would be better optimised for each other. This could align the two-tiered results more closely to work in the literature, but was unfortunately not achievable with the tools used in this project. The LSTM also under performed compared to expectations. It was expected that due to its ability to take into account multiple timesteps, and therefore time dependent behaviours, it would have an

advantage over the other models. However, it still only performs similarly. The lacklustre LSTM performance could be due to the relatively low frequency of the input data. The 30 ms intervals are much longer then several important timescales; thus, information that occurs on these scales cannot be recovered. Moreno et al. [35] also identified a low temporal resolution as one of the causes for incorrect classifications in the APODIS system in JET. Additionally, the high noise on the data, similar to that observed in Figure 5, could negate the algorithms' ability to identify clear patterns in the data.

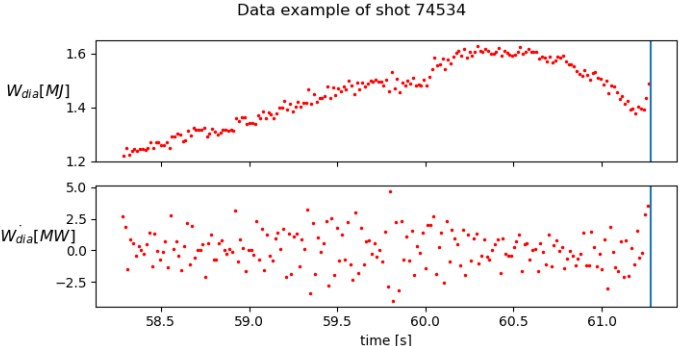

**Figure 5.** Example of diamagnetic energy and its time derivative, showing the noise on the signals. The vertical line is the moment of disruption.

Figure 6 shows dimensionless predictor performance metrics. There is an overall decrease in performance across the board. This can most likely be attributed to the loss in information in the transition from standard to dimensionless inputs. In the latter, the 10 original inputs are combined into only 6 features. However, it can be observed that not all ML methods are equally sensitive to this transition. SVM and LSTM, which had the best performance in the standard models, had a drop of around 0.15 in the F1-score. The two-tiered model only dropped about 0.10. GBT and RF experienced the lowest impact with a drop of only 0.05. This makes RF the best performing among the dimensionless models. We cannot offer a clear explanation for the different sensitivities, but empirically this suggests an advantage for RF and GBT for the design of the dimensionless models. To more comprehensively test the effective portability of different models, they should be tested on different devices, as was for example conducted in Rea et al. [13], but this was outside the scope of this work.

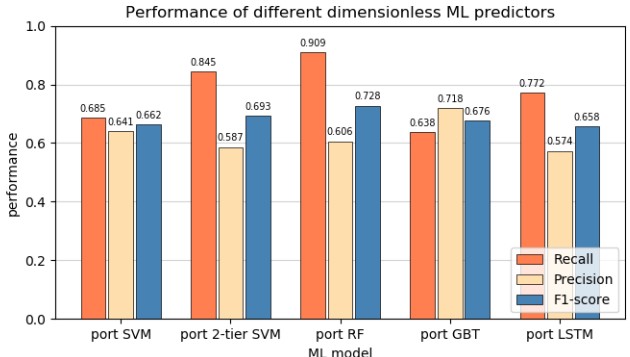

**Figure 6.** Recall, precision and F1-score of the different dimensionless prediction models.

### 4.2. Detection Time and Cumulative Detection Rate

Figures 7 and 8 show the evolution in time of the detection rate before a disruption for, respectively, the standard and dimensionless models. The detection rate is the ratio of detected disruptions over the total disruptions. It therefore indicates when disruptions are detected for the first time in the 600 ms disruptive samples. The right-most value in these graphs corresponds to the total detection rate, which is the same as the reported

recall. For all the models, the detection rate starts at a high value. This indicates that a significant fraction of the disruptions are already detectable at 600 ms before the disruption, suggesting that in future work, a longer time window before the disruption should be considered. This seems to be confirmed by comparing to Figure 5 of López et al. [38], where defections in JET data are measured as early as 2 s in advance. Of course, this does not necessarily justify increasing the disruptive time scale, as this would also increase the false alarms. The high initial detection rate can also be inflated by false alarms. If the false alarm rate is high, the model predicts a high amount of disruptions, even when they are not imminent. This high amount of unfounded detections would also happen close to the disruptions, but would in this case be labelled as a correct detection and thus create a background detection rate. The false alarm rates of the different models are listed in Table 3. A high false alarm rate indeed corresponds to a higher initial detection rate. Nonetheless, the false alarm rates are not large enough to fully explain this on its own. Notice that the two-tiered models start at −540 ms. This is due to the sliding window scheme, requiring three datapoints to work.

**Table 3.** False alarm rate for the different models.

|               | SVM   | T2    | RF    | GBT   | LSTM  |
|---------------|-------|-------|-------|-------|-------|
| Standard      | 0.136 | 0.155 | 0.223 | 0.123 | 0.149 |
| Dimensionless | 0.189 | 0.294 | 0.291 | 0.123 | 0.283 |

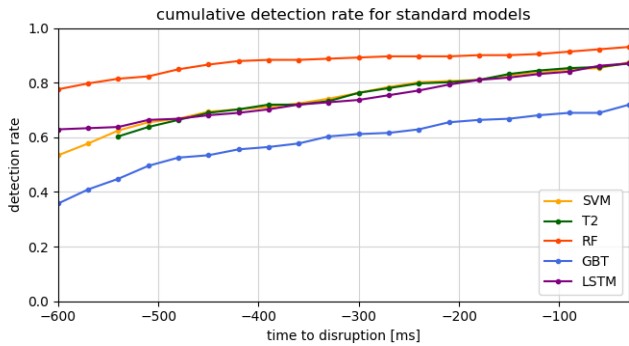

**Figure 7.** Cumulative detection rate for the different standard predictor models.

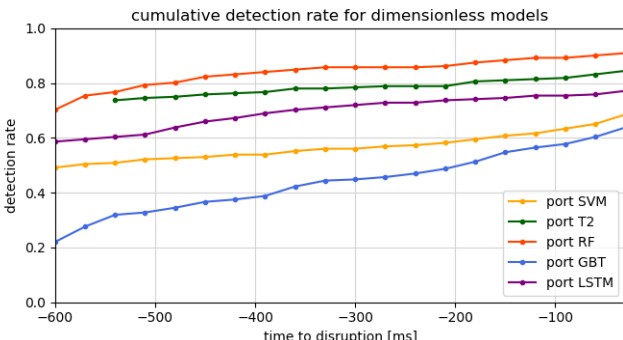

**Figure 8.** Cumulative detection rate for the different dimensionless predictor models.

### 4.3. Relative Feature Importance

The RF method has been used to determine the relative importance of the different input features, as described in Breiman [20] and Louppe et al. [39]. Such information can be very valuable in designing future predictors by being able to detect important features or remove less important ones for performance optimisation. Note that the RF method is certainly not the only way of determining the feature importance, but other methods have been left for future work. The results are shown in Figure 1 for the standard models.

Overall, the errors are quite high, disallowing clear cut conclusions. The exceptions are the time derivative of the diamagnetic energy and the NBI and ICRH input powers, which are all deemed significantly less important by the RF algorithm. The time derivative of the diamagnetic energy has the lowest importance of all inputs. This could be due to the noise on the diamagnetic energy signal, which can be observed in Figure 5. This makes for a very erratic time derivative. This could be overcome in the future work by smoothing the time derivative, though this comes at the cost of physical interpretability, due to the quantity no longer being a true derivative. The ICRH and NBI input power lower performance is likely caused by the frequent absence of this data in the database. However, a similar decreased performance for the internal plasma inductance, which is missing in 18.7% of the data, cannot be observed. This could be due to these data being missing far less frequently or the reduced importance being offset by a high underlying physical importance of the quantity. Figure 9 contains the feature importance's for the dimensionless models. Here, the radiated power fraction $f_P$ has the lowest importance. $f_P$ depends on $\dot{W}_{dia}$ and $P_{in} = P_{NBI} + P_{ICRH}$, which have the lowest importance in the standard model as well. This effect could be amplified by the large loss of information due to combining four quantities into one, as was discussed in Tang et al. [28]. In this paper, it also suggested that the mode lock fraction is a poor choice as input feature, due to its lower observed performance. The internal inductance appears to have lower performance as well, potentially caused by its frequent unavailability in the dataset. This suggests the need for further research into the reasons for poor behaviour and the construction of better portable features, which could better retain all the information contained in the original feature set.

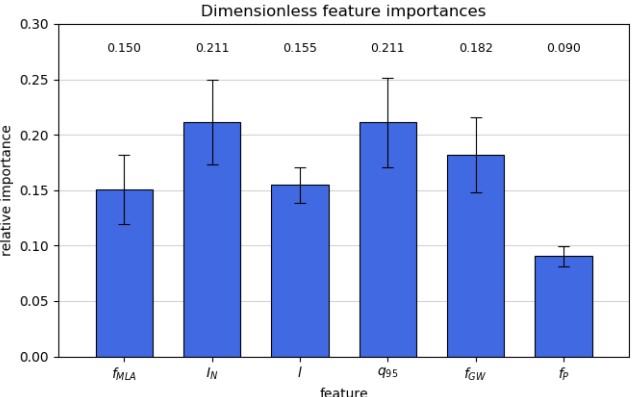

**Figure 9.** Relative importance of the different dimensionless input features, as obtained from the random forest model.

### 4.4. Computational Cost

Table 4 shows both train and inference time for the different models. The train time here means the time required to train a single model using a given set of hyperparameters, and therefore does not include the hyperparameter fitting. The inference time is the average time required to make a new prediction, given an input. This is important because slow inference time could leave insufficient time for disruption mitigation systems to be activated. These values are hardware dependent. In this project, an intel i7-7700HQ at 2.8 GHz was used. The LSTM model also used an Nvidia GTX 1050ti 4GB GPU as the accelerator during training, significantly speeding up the training time to values similar to the other models. GBT and SVM are the fastest in terms of training time, followed by RF, LSTM and lastly the two-tiered SVM. The first tier of this model needs to output the probabilities of disruption rather than discrete classifications, which significantly increases the training time. GBT has the fastest inference time, followed by LSTM and RF. Finally, there is SVM and the two-tiered SVM. The tiered model is the slowest, since it requires two SVMs to be evaluated. However, the second tier SVM is much simpler and therefore only slightly increases the compute time over the standard SVM. Given that the slowest model, the two-tiered SVM,

is being used as a disruption detector in JET [38], it can be assumed that all models are fit for disruption prediction in terms of inference time.

**Table 4.** Train and inference time information for the different models. (* LSTM was trained on a GPU to speed up computation).

|  | SVM | T2 | RF | GBT | LSTM |
|---|---|---|---|---|---|
| Train time [s] | 28.21 | 184.89 | 138.52 | 24.88 | 79.18 * |
| Inference time [ms] | 0.43 | 0.52 | 0.11 | 0.007 | 0.05 |

## 5. Conclusions

Standard and dimensionless disruption prediction models for JET have been designed using five different machine learning techniques. In the standard models, the performances, as measured by the F1-scores, are very similar. This could indicate an underlying issue with the dataset used, potentially due to choices made in its setup in this project. Careful study of the incorrectly predicted samples could help identify such issues to prevent them in future work. Frequent data gaps are also a likely cause for reduced performance, as they negatively impact feature importance and were identified in the literature as a cause for incorrect predictions [35]. Carefully designed systems to fill in data gaps such as Rattá et al. [36] could therefore prove invaluable. Nonetheless, SVM has an overall slight advantage over the other models. Particularly remarkable is the slightly lower performance of the two-tiered SVM, since it is so similar to the standard SVM, and should in theory have an advantage. This could be attributed to the inability to propagate errors between tiers in this model. Also significant is the unremarkable performance of the LSTM. It was expected that this algorithm's ability to analyse time-dependent inputs would give it a clear edge in this comparative study. This was, however, not observed, potentially due to the low frequencies and high noise of the input signals, severely limiting the temporal information that can be retrieved. The negative impact of the noise was also identified by the feature importance analysis and Moreno et al. [35]. This could suggest that careful data preparation to reduce the noise could improve the performance for such models in future work, though a proper analysis into such techniques is required. The analysis of the time evolution of the recall in disruptive samples shows a very high recall at 600 ms before the disruption. As the false alarm rate cannot explain this, it suggests that a longer disruptive window should be considered in future work with JET data, which also corresponds with the literature López et al. [38]. However, this could also increase the mislabelling of the non-disruptive data points and therefore decrease the performance. A proper analysis of the disruptive time window and its impact on the different models should therefore be conducted in future work. In transitioning to dimensionless parameters, a drop in performance is observed across the board, as is expected due to the loss in information in this transition by going from 11 to 6 features. Feature importance analysis supports findings from Tang et al. [28], though not conclusively, suggesting tha mode lock fraction and radiated power fraction are poor replacement features, and alternatives should be considered in future work. However, RF and GBT demonstrated a demonstrably lower decrease in performance due to the transition, making RF the best performing dimensionless model, and the further investigation of this method is left for future work to rigorously verify these findings. Such portable models should also consider the change in relative disruptive domains between devices, which is expected to further reduce the performance [29]. It would be interesting to examine whether RF and GBT are also more robust against this effect. This could be tested in future work by applying these methods to data from different tokamak devices.

**Author Contributions:** J.C. conducted the development and training of the ML tools under the guidance of J.A. and analysed the results. G.L. contributed to the itnerpretaton of the results. The manuscript was prepared by J.C. with supervision from G.L. and J.A. All authors have read and agreed to the published version of the manuscript.

**Funding:** This research was funded by Onderzoekfonds KU Leuven (project C14/19/089).

**Institutional Review Board Statement:** Not applicable.

**Informed Consent Statement:** Not applicable.

**Data Availability Statement:** JET database.

**Acknowledgments:** The authors would like to thank EUROfusion and CCFE for access to the JET database. Special thanks to Dirk Van Eester from LPP-ERM/KMS for his support and invaluable help in accessing and using this database. The work reported here received support from the Onderzoekfonds KU Leuven (project C14/19/089). This research used resources of the National Energy Research Scientific Computing Center, which is supported by the Office of Science of the US Department of Energy under Contract No. DE-AC02-05CH11231.

**Conflicts of Interest:** The authors declare no conflict of interest.

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
