# Peer review of "Investigation of Machine Learning Techniques for Disruption Prediction Using JET Data"

_plasma, doi:10.3390/plasma6010008_

Round 1

Reviewer 1 Report

Investigation of machine learning techniques for disruption 2 prediction using JET data

J. Croonen joost.croonen@kuleuven.be, J. Amaya and G. Lapenta Centre for mathematical Plasma Astrophysics, KU Leuven, Leuven, 3000, Belgium

Abstract:

The abstract covers and presents the main steps of this research. Its volume is sufficient, but if the authors decide, they can easily fill it with other important points of interest in their conclusions.

Keywords:

No keywords were selected and provided.

Main text:

Congratulations to the authors for this undeniably useful work. That the direction of the specific research is in a field with extraordinary technologies, but this approach in a more "primary" form is used to solve multiparametric problems, and applications can be found in all engineering fields. The main argument is the construction of sustainable and durability systems. The innovation here can be found in the fine-tuning of tree structures with the empirical determination of priority branches through the data system after proper noise cancellation.

Paragraph 1

In the research and argumentation of the necessity and importance of this research, some studies with a purely analytical direction can be found, such as:

• PUSTOVITOV, V. D.; RYABUSHEV, E. A. Effect of Pressure Anisotropy on Diamagnetic Signal in a Tokamak with Noncircular Plasma Cross Section. Plasma Physics Reports, 2021, 47.10: 947-955.

• SCHLISIO, Georg. Analysis of the gas balance for Wendelstein 7-X. 2021,

           and other.

I recommend that the abbreviations used be explained and described through analysis.

A comment was made in Table. 1 that the parameters based on Ratt´a et al. (2012) and Rea et al. (2018) are not fully comprehensive, and in that case it may be appropriate to discuss which others would be useful for improving the models.

Paragraph 2

An explanation of the abbreviation is useful in their first occurrence of said abbreviation.

The comment in the first paragraph is interesting. Granting the equivalence of factors requires explanation and study, and the events mentioned may fall into the category of design accident. The conclusions about the engineering treatment of the problems should be considered with particular weight assigned to it.

Line 64 – Ratt´a et al approach is commented. (2012) and Rea et al. (2018), it would be interesting to compare the approach used by the authors and the citations. If the authors consider they can present at least an analysis and their conclusions from this study.

Line 65 – If the normalization criterion is one standard deviation, what is the value of the variation?

Line 71 – Has the approach to data been carefully evaluated from a statistical point of view? This requires evaluation especially in cases where the data is of one character.

Paragraph 80-92 – The comments on the cited Ratt´a et al. (2012) and Rea et al. (2018) are particularly important as they report a result. Here the question arises whether temporal pixelation (or duration) or other markers can be found to serve as predictive indicators. This would be an important marker for this study as well, as a verification marker.

Paragraph 93-105 – As the authors point out the choice of a conservatively wide time interval (this is disputed in the conclusion) allows the problem to be solved without some strict constraints. However, the point of interest here is the expansion of this interval and its impact on the forecast. However, the authors claim that this is not in their direction of  interest. In my opinion, when comparing methods, the assessment of the level of credibility is important and this can be reconsidered.

Параграф 3

Paragraph 120-123 – An important choice, but to it can be added an approach to extend the test and verification sets to evaluate these influences as well.

An appropriate presentation of the compared algorithms is an important step in this research.

Параграф 4

Lines 201 -205 - The presented formulas should be numbered. In the formula of line 205, the two evaluation factors R and P are assigned the same priorities. Is this not a skewing the result assumption? This is commented on briefly, but can be developed with the results of Figure 3.

Line 221 - Your comment here is important and I ask that you link it to my earlier comments.

Line 228 - Interesting comment on the SVM model. Can criteria for sufficiency of performance improvement steps be identified?

Paragraph 266 -283 – The conclusion drawn here is important and clear reasons should be sought and solutions recommended (perhaps in the authors' future work).

Conclusion

The important conclusion of line 313 about the influence of noise also requires a solution. Couldn't this be done by looking for a setting in the normalization?

My comments above are supported by the authors' conclusions on line 317, and this should be set aside as a declarative intent for future research.

The conclusion of line 326 should, in my opinion, be reconsidered.

I hope that my comments will help the authors improve this already useful study.

 Please review the format and meet the release requirements.

Author Response

We thank the referee for the useful and constructive comments. Below we report our responses in the order followed by the referee. We have followed the suggestion and augmented the abstract with additional useful information about the content of the paper . Keywords were added. Paragraph 1 We have indeed found the references quite interesting and added them. We have included all acronyms as suggested. We have added a discussion of the other parameters that could be of interest. Paragraph 2 We have again added the abbreviation explanation. As for the engineering treatment of the problems we refer to the literature as we think the subject is not directly part of our narrative. Line 64: We have added a short discussion on the different methods used in the two papers, to highlight their approaches. Line 65: The relative standard deviations have been added to the input feature table. Line 71: No in depth data analysis has been done to identify the influence of this data processing method. Since the focus lies on comparing the models rather than building an operational model, it was chosen to be acceptable, since all models still work on the same data sets, such that the comparison still stands. Paragraph 93-105: We agree that the topic is of interest but we have not addressed it in our study and we refer the reader to our conclusions and future work for this aspect of further research. Paragraph 3: As for point 11, we added a sentence to suggest this comment for future work. Paragraph 4: Lines 201- 205, we have now numbered the equations. We have also added a discussion on the possible alternatives to giving R and P the same priority. Line 221- We added a note on the importance of generated data and linked it to the previous comment. Line 228 - The criteria for improved 2-tier SVM performance is the ability to simultaneously train the tiers together with proper error-propagation between them, but this was not achievable with the tools used in this project. The text has been adjusted to more clearly explain this. Line 266-283. As suggested by the referee we added this comment in the future work section. Conclusion: The noise issue in line 313 indeed can be improved using pre-treatment of the data. We added a comment on this. We further clarified the comment in line 317 Conclusion in line 326 has been removed.

Reviewer 2 Report

In the present manuscript, the authors proposed machine learning techniques for disruption prediction using JET data. For this, the author's employed a support vector machine, a 2-tiered support vector machine, random forest, gradient-boost trees, and long-short term memory ML models. Before acceptance of the manuscript, the authors need to address the following comments:-

1) Page 1, line 23, "As machines are scaled to reactor sizes, the damage such events can cause can be catastrophic." The sentence seems to be incorrect/misleading. A careful revision is required by authors to avoid such typos/errors. 

2) Authors must include and cite the latest state of art methodologies in similar fields. 

3) The introduction section needs complete revision. Authors must focus on Mitigating/plasma disruptions, their cause etc.

4) Page 2, line 43, "An important but not sufficient condition for portability is a transition to dimensionless quantities." Clearly signify the statement. 

5) From Table 1, the predictors, "plasma internal inductance", "Neutral beam injection input power [W]" and "Ion cyclotron radio heating input power" has nearly 20%, 30%, and 58% missing values respectively. The authors replaced the missing values with zero. Certainly, this will induce errors in the prediction performance. Imputation techniques such as K-NN missing imputation technique, mean value imputation, etc. can be applied to improve the performance. 

6) Fig 8 and 9 require more explanation for better understanding.

Author Response

We have changed the text to be less hyperbolic and more accurately state the consequences of disruptions. Several additional references have been added of recent developments in the field. The introduction has been expanded upon with the aforementioned additional references, and information on both disruption mechanisms and mitigation methods, though these discussions have been kept surface level as they are neither our field of expertise nor the focus of the paper, and references have been provided for interested readers to further investigate these topics. This statement has been expanded and emphasised. This is an interesting idea, and will be considered for future developments. This can be considered alongside the regression methods suggested by Ratta et al 2014. The section discussing these figures has been expanded and additional references were added for further explanation.

Round 2

Reviewer 2 Report

All my quires were replied satisfactorily.